# Androgen Receptor Is Expressed in the Majority of Breast Cancer Brain Metastases and Is Subtype-Dependent

**DOI:** 10.3390/cancers15102748

**Published:** 2023-05-13

**Authors:** Kevin Yijun Fan, Rania Chehade, Maleeha Qazi, Veronika Moravan, Sharon Nofech-Mozes, Katarzyna J. Jerzak

**Affiliations:** 1Faculty of Medicine, University of Toronto, Toronto, ON M5S1A8, Canada; 2Department of Medical Oncology, Faculty of Medicine, University of Toronto, Toronto, ON M5S1A8, Canada; 3VM Stats, Toronto, ON M5A4R3, Canada; 4Department of Laboratory Medicine and Pathobiology, University of Toronto, Toronto, ON M5G1X5, Canada; 5Sunnybrook Odette Cancer Centre, University of Toronto, Toronto, ON M4N3M5, Canada

**Keywords:** androgen receptor, breast cancer, brain metastasis, immunohistochemistry

## Abstract

**Simple Summary:**

Androgen receptor (AR) is a receptor found on many breast cancer cells, and drugs can be used to target AR in the treatment of advanced breast cancers. We aim to better understand how AR is expressed in breast cancer that has spread to the brain, i.e., brain metastases (BrM). We used a technique called immunohistochemistry to study AR expression in BrM from 57 breast cancer patients. We found that AR was expressed in the majority of BrM, and was expressed at different frequencies across different types of breast cancer. We did not find an association between AR expression and measures of survival. In most patients, AR expression in breast tumours and BrM was comparable. This study shows that AR is expressed in the majority of breast cancer BrM, and may be a useful target for treating breast cancer patients who have BrM.

**Abstract:**

We aimed to evaluate the expression of the “targetable” androgen receptor (AR) in breast cancer brain metastases (BrM). An established, retrospective 57-patient cohort with metastatic breast cancer who underwent surgery for BrM at the Sunnybrook Odette Cancer Centre between 1999–2013 was studied. AR expression in BrM samples was assessed in triplicate using immunohistochemistry (IHC). AR positive status was defined as nuclear AR expression ≥ 10% by IHC using the SP107 antibody. The median age of patients was 52 years (range 32–85 years). 28 (49%) of BrM were HER2+, 17 (30%) were hormone receptor positive (HR+)/HER2−, and 12 (21%) were triple negative breast cancers (TNBCs). 56% (n = 32/57) of BrM were AR positive, and median AR expression was 20% (CI 1.6–38.3%). AR expression was different across breast cancer subtypes; AR was most frequently expressed in HER2+ (n = 21/28), followed by HR+/HER2− (n = 9/17), and lowest in TNBC (n = 2/12) BrM (*p* = 0.003). Patients with AR positive versus AR negative BrM had similar overall survival (12.5 vs. 7.9 months, *p* = 0.6), brain-specific progression-free survival (8.0 vs. 5.1 months, *p* = 0.95), and time from breast cancer diagnosis to BrM diagnosis (51 vs. 29 months, *p* = 0.16). AR is expressed in the majority of breast cancer BrM and represents a potential therapeutic target.

## 1. Introduction

Androgen receptor (AR) is a nuclear steroid hormone receptor that functions as a ligand-activated transcription factor and regulates the expression of downstream genes involved in cellular proliferation and apoptosis [1,2]. AR is ubiquitously expressed in 72–79% of primary breast cancers [3,4,5], although its expression differs by breast cancer subtype, and is more frequently expressed in hormone receptor positive breast cancer compared to hormone receptor negative breast cancer [5,6]. Triple-negative breast cancer (TNBC) is traditionally thought to be devoid of molecular targets, yet AR is expressed in 27–35% of cases [3,5,7]. As a result, several phase II trials have evaluated the safety and efficacy of androgen-targeted therapies, including abiraterone, enzalutamide and bicalutamide, among patients with metastatic TNBC with clinical benefit rates ranging from 19–35% [8,9,10,11,12]. The safety and efficacy of androgen-targeted therapies with standard of care systemic therapies, such as trastuzumab for patients with advanced HER2+ breast cancer [13], and exemestane for those with hormone receptor (HR) positive disease, have also been evaluated [14].

Given that breast cancer is the second most common cause of brain metastases (BrM), which are associated with high rates of morbidity and mortality, there is interest in evaluating novel treatment approaches with central nervous system (CNS) activity [15]. The anti-androgen agents enzalutamide and apalutamide are known to cross the blood-brain-barrier and hold promise as targeted treatment options for patients with AR-positive metastatic disease [16]. However, little is currently known about the expression of AR in breast cancer BrM [17]. 

We aimed to evaluate the expression of AR using a clinically validated and commercially available antibody in an established cohort of patients with metastatic breast cancer who underwent surgery for BrM at our institution. 

## 2. Materials and Methods

### 2.1. Study Population

This study used a retrospective cohort of breast cancer patients who underwent surgery for BrM at Sunnybrook Odette Cancer Centre (SOCC) in Toronto, Ontario, Canada, consecutively between July 1999 and June 2013. The Anatomic Pathology Laboratory Information System was searched for metastatic carcinoma of breast origin. Patients with a personal history of previous or concurrent breast carcinoma and BrM and no evidence of another primary tumor were included if their breast primary pathology was available in the archives of the Pathology Department. Clinical information as well as information on BrM and matched primary breast tumours were obtained from the patients’ electronic medical records. Mean followup time was 11.4 months. Research Ethics Board approval was obtained from the Sunnybrook Research Institute.

### 2.2. Immunohistochemistry

Immunohistochemistry (IHC) was assessed on BrM samples using an established tissue microarray (TMA) consisting of 1 um tissue cores in triplicates. AR expression was assessed using the SP107 antibody (Sigma-Aldrich (St. Louis, MO, USA)) and was scored as a continuous variable as percentage of positive nuclei. Positive status was defined as nuclear AR expression ≥10% of tumor cells. GATA3 expression was assessed using the sc-268 antibody (Santa Cruz Biotech (Dallas, TX, USA)) and Ki-67 expression was assessed using the clone 30-9 antibody (Ventana (Tucson, AZ, USA)). Both GATA3 and Ki-67 expression were scored as a continuous variable as percentage of positive nuclei, and we defined low expression as 1–24%, intermediate expression as 25–49%, and high expression as ≥50%. Estrogen receptor (ER), progesterone receptor (PR), and human epidermal growth factor receptor (HER2) status of the BrM was evaluated based on 2020 and 2018 ASCO/CAP guidelines respectively [18]. Five patients had two resected BrM; in these cases we took the average AR expression of the two BrM. AR expression was also evaluated in the matched primary breast tumours for 10 patients using similar methods. 

### 2.3. Statistical Analysis

Brain-specific progression-free survival (bsPFS), OS, and time from diagnosis of breast cancer to diagnosis of BrM were estimated using the Kaplan–Meier method and compared across groups using the log-rank test. Independence between categorical variables was assessed using the χ2 test. Mean AR expression across breast cancer subtypes was compared using ANOVA, and post-hoc analysis using Tukey HSD. Difference between AR expression in primary breast samples and matched BrM was tested using the Wilcoxon signed rank test. bsPFS was defined as the duration of time in months from the time of BrM surgery to the time of BrM progression or death. Overall survival (OS) was defied as the duration of time in months from the time of BrM surgery to the time of death due to any cause. Time from diagnosis of breast cancer to diagnosis of BrM was defined as the duration of time in months from first diagnosis of primary breast cancer to the time of BrM surgery. For all analyses, statistical significance was defined as a *p*-value of <0.05. Data was analyzed using R version 4.2.1 (23 June 2022) and visualized using the ggplot2 package.

### 2.4. Systematized Review of Active Clinical Trials Investigating AR Targeted Therapies in Advanced and Metastatic Breast Cancer

A search was conducted on 25 January 2023 on the clinicaltrials.gov website using the search terms “Breast cancer” and “Androgen” and filtered for studies that were “Recruiting”, “Not yet recruiting” or “Active not recruiting”. This search yielded 37 trials, which were manually reviewed. Only trials including women ≥18 years with advanced or metastatic breast cancer, with the primary intervention being an androgen-targeted therapy either alone or in combination with other systemic therapies were selected for review. 16 trials met the inclusion criteria and are included in Table 1.

## 3. Results

### 3.1. Baseline Patient Characteristics

57 patients with metastatic breast cancer who underwent surgical resection of BrM were included in this study (Table 2). The median patient age at the time of BrM diagnosis was 52 years (range 32–85 years, IQR 14 years). 28 (49%) of BrM were HER2+, 17 (30%) were hormone receptor positive (HR+)/HER2−, and 12 (21%) were TNBC. 61.4% (n = 35) of patients had a single BrM and the median size of BrM was 3 cm (range 0.3 cm to 6.2 cm, IQR 1.4 cm). The BrM were resected from the following locations: cerebellar (43.9%, n = 25), frontal (26.3%, n = 15), parietal (22.8%, n = 13), temporal (5.3%, n = 3), and occipital (1.8%, n = 1) regions. Tumor grade of BrM was available for 51 (89%) cases: 28.1% (n = 16) were grade 1, 35.1% (n = 20) were grade 2, and 26.3% (n = 15) were grade 3. The majority (87.7%, n = 50) of patients had neurological symptoms at the time of presentation with BrM. The most common sites of extra cranial metastases were bone (38.6%, n = 22), lung (29.8%, n = 17), liver (24.6%, n = 14), lymph node (19.3%, n = 11), and chest wall (3.5%, n = 2). After surgical resection of BrM, 89.5% (n = 51) of patients underwent adjuvant radiation therapy. Information on systemic therapy prior to BrM development was available for 23 patients, of whom 17.5% (n = 10) of patients received HER2-targeted therapy, 15.8% (n = 9) received chemotherapy, and 7.0% (n = 4) received endocrine therapy; no patients received androgen-targeted therapy.

### 3.2. AR Expression in BrM

AR expression by IHC using the SP107 antibody was quantified for all 57 BrM, and is shown in Figure 1a. Median AR expression level was 20.0% (IQR 87.5%), and mean AR expression was 40.1% (SD 41.4%). Among 57 BrM, 20 BrM lacked AR expression entirely (i.e., had 0% AR expression by IHC; this included 10 TNBC, 7 HR+/HER2−, and 3 HER2+ BrM) and 5 BrM expressed low levels of AR (i.e., between 1–9% by IHC). The remaining BrM had AR expression between 10–49% (n = 6) or 50–100% (n = 26).

In subsequent analyses, cases with AR expression ≥10% were labelled as “AR positive (AR+)”, while those with AR expression <10% were labelled as “AR negative (AR−)”. This was based on a previous study that found that AR expression threshold of 10% is associated with clinical response to enzalutamide in TNBC [8]. 56% (n = 32/57) of BrM were AR positive. Distribution of BrM subtype, Ki-67 expression, GATA3 expression, age at BrM diagnosis, BrM location, BrM size, and BrM grade, as stratified by AR status, is shown in Table 3. No significant associations were found between AR status and age at BrM diagnosis, BrM location, BrM size, or grade of BrM (Table 3).

### 3.3. AR Expression by Breast Cancer Subtype

We found a significant association between AR expression and primary breast cancer subtype. AR was most frequently expressed in BrM from HER2+ breast cancer (n = 21/28 BrM were AR+), followed by HR+/HER2− breast cancer (n = 9/17), and least frequently in TNBC (n = 2/12) (*p* = 0.003) (Figure 1b). Mean absolute AR expression was also highest in BrM from HER2+ breast cancer (mean AR expression 53.1%), followed by HR+/HER2− breast cancer (36.5%), and lowest in TNBC (15%) (ANOVA *p* = 0.023, *p* = 0.019 for Tukey HSD between HER2+ vs. TNBC) (Figure 1c).

### 3.4. AR Expression in Relation to Ki-67 Expression and GATA3 Expression

We found a significant association between Ki-67 expression by IHC and both frequency of AR expression (Figure 2a, *p* = 0.020) and mean absolute AR expression (Figure 2b, *p* = 0.005). The subgroup with high (50% or above) Ki-67 expression had the lowest AR expression (*p* = 0.005 for Tukey HSD between Ki-67 high vs. Ki-67 intermediate groups) (Figure 2b).

We also found a significant association between GATA3 expression by IHC and both frequency of AR expression (Figure 2c, *p* = 0.003) and mean absolute AR expression (Figure 2d, *p* = 0.005). The subgroup with high (50% or above) GATA3 expression had the highest AR expression (*p* = 0.017 for Tukey HSD between GATA3 high vs. GATA3 negative groups, *p* = 0.035 for Tukey HSD between GATA3 high vs. GATA3 low groups) (Figure 2d).

### 3.5. AR Expression in Matched Primary Breast Cancers

In 10 patients for whom matched BrM and primary breast cancers were available, AR expression by IHC was higher in primary breast cancers compared to matched BrM (79% vs. 52%, Wilcoxon signed rank test *p* < 0.05). 90% (n = 9/10) of primary breast cancers were AR+, whereas 70% (n = 7/10) of BrM were AR+. AR status was concordant in 7 of 10 cases. Among the 3 cases for which AR status differed in the primary cancer and BrM, AR was “gained” in the BrM in one case and “lost” in the BrM in two cases (Figure 3). 

### 3.6. Associations between AR Expression and Clinical Outcomes

Patients with AR+ BrM had a median OS of 12.5 months, compared to patients with AR− BrM who had a median OS of 7.9 months (*p* = 0.6) (Figure 4a). Patients with AR+ BrM had a median bsPFS of 8.0 months, compared to patients with AR− BrM who had a median bsPFS of 5.1 months (*p* = 0.95) (Figure 4b). These differences were not statistically significant.

Patients with AR+ BrM had a numerically longer median time from diagnosis of breast cancer to diagnosis of BrM, compared to those with AR− BrM, although this was not statistically significant (51 vs. 29 months, *p* = 0.16) (Figure 4c).

## 4. Discussion

AR expression in breast cancer BrM is of interest due to readily available androgen-targeted therapies that can cross the blood brain barrier. This study demonstrated in a retrospective Canadian cohort that AR is expressed by IHC in the majority of breast cancer BrM, and that HER2+ BrM most frequently expressed AR, while TNBC BrM least frequently expressed AR. AR status was concordant between the majority of BrM and matched primary breast tumours. AR status of the BrM was not associated with OS or bsPFS, although patients with AR+ BrM had a trend towards developing BrM later in the course of their disease. 

In our study, AR expression by IHC was expressed (i.e., expression by IHC > 0% using SP107) in 64.9% of breast cancer BrM. This is higher than what was reported in a previous study which demonstrated that AR expression was expressed (i.e., expression by IHC > 0% using AR441) in 35.1% of breast cancer BrM in a similar-sized cohort [17]. The antibody used in our study (SP107) has been shown to have higher sensitivity and robustness compared to AR441 [8]. In our analyses, we also used a clinically relevant AR threshold of ≥10% expression, which corresponds to the threshold associated with clinical response to enzalutamide in TNBC [8]. 

With respect to AR expression by IHC in primary breast cancers, our results are comparable to those of other studies, which consistently report that AR is expressed in 72–79% of primary breast cancers [3,4,5]. In our analyses of BrM, AR is most frequently expressed in HER2+ BrM (previously reported to be expressed in 76–87% of primary HER2+ breast cancers [3,19,20]), and AR is least frequently expressed in TNBC BrM (previously reported to be expressed in 27–35% of primary TNBC) [3,5,7]. The largest among these studies, which was performed on 2171 patients in the US between 1976 and 1996 using the AR441 antibody, reported AR expression (≥1%) in 91% of Luminal A, 68% of Luminal B, 59% of HER2-postive, and 32% of triple negative primary breast cancers [5]. We observed a similar pattern of AR gain and loss between primary breast cancers and matched metastases as a previous study, which showed that AR was expressed ≥10% by SP107 in 82.9% of 164 primary breast tumours and 60.2% of 83 metastases, with a concordance rate of 60.6% between primary breast cancers and matched metastases [21]. The variability in concordance between the AR status of primary breast cancers and matched metastases suggests that re-checking AR status in metastatic sites may be of value when considering the use of an AR-targeted therapy. 

While AR expression by IHC is the gold standard, previous studies have also investigated mRNA expression of AR in primary breast cancer, although to our knowledge there have been no such studies in breast cancer BrM. A study of 101 primary TNBCs found that qRT-PCR detected AR expression in 34% of cases compared to 15% of cases using IHC (i.e., ≥1% using AR441 antibody), with 75% concordance between the two methods; this study found that AR mRNA expression was associated with shorter distant metastases free survival, which is consistent with previous studies employing IHC in TNBC [22]. Another study of 872 primary ER-negative breast cancers found that mRNA expression of AR had a strong correlation (R = 0.68, *p* < 0.001) with AR reverse phase protein assay, and AR mRNA expression was associated with lower grade disease and lower risk of recurrence [23]. Interestingly, this study also showed that AR mRNA expression was associated with HER2+ status, which is consistent with our finding that HER2+ BrM have highest expression of AR by IHC. While qRT-PCR may be a more sensitive method that can identify patients whose tumours have low levels of AR expression not detected by IHC, the likelihood that such patients would benefit from androgen-directed therapies is uncertain, given that AR IHC expression thresholds for clinical response to androgen-directed therapies are relatively high, e.g., ≥10% for clinical response to enzalutamide in TNBC [8], and ≥40% for clinical response to enobosarm in HR+ breast cancer [24]. Presently, IHC remains the most extensively studied and most clinically relevant method for assessing AR expression.

Our study found no significant association between AR expression with either OS or bsPFS, which is not surprising given that our cohort consisted of multiple different breast cancer subtypes, which inherently portend different prognoses; the number of patients within each subtype was too small to draw meaningful conclusions. Additionally, no patients in our study received AR-targeted therapies. 

AR signaling plays a different biological role in different breast cancer types [11]. In ER+ breast cancer, previous studies have shown that positive AR status was associated with longer breast-specific survival [25,26]. This is not surprising given that AR functions as a tumour suppressor in ER+ breast cancer. The activation of AR represses ER-regulated cell cycle genes and upregulates AR target genes which include tumour suppressors [27], and this supports the rationale for ongoing trials evaluating AR agonists in ER+ breast cancer (Table 3). In TNBC, the prognostic significance of AR is more controversial [28,29,30,31,32]. In TNBC, AR binds to androgen-related element in the nucleus to induce cell proliferation [33,34]. This supports the role for ongoing trials evaluating AR antagonists in TNBC (Table 3), with the caveat that the utilization of robust biomarkers for optimal patient selection is paramount. For example, luminal AR+ TNBCs express high levels of AR and are particularly sensitive to bicalutamide in preclinical models [35]. Finally, in HER2+ breast cancer, enzalutamide inhibited growth in trastuzumab-resistant HER2+ breast cancer xenografts [36], which may be attributed to the cross-talk between AR signaling and HER2 signaling [34]. A recent phase II clinical trial demonstrated that enzalutamide plus trastuzumab was well tolerated in patients with AR+/HER2+ breast cancer who were pretreated with anti-HER2 therapy, and a subset of these patients had durable disease control [13]. However the median PFS was only 3.4 months and patients with BrM were not included in this trial [13]. Our study showed that HER2+ breast cancer BrM most frequently express AR, and suggests that there is a need to elucidate the biology of AR in HER2+ breast cancer. 

Many of the associations that we found between the expression of AR and the expression of other prognostic and proliferation markers (GATA3 and Ki-67) are in keeping with what is known about the biology of breast carcinomas. Ki-67 is a nuclear protein associated with cellular proliferation, and is associated with higher histological grade and shorter survival [37]. The fact that we did not find a linear relationship between Ki-67 and AR expression is not surprising, given that AR plays opposing roles in proliferation among different breast cancer subtypes. The subgroup with high (50% or above) Ki-67 expression had the lowest AR expression, and the majority (5 of 9) of BrM which were Ki-67 high/AR low were TNBC. This is consistent with the fact that TNBC has a higher rate of proliferation and less frequent expression of AR. GATA3 is a transcription factor that plays a role in the differentiation of breast luminal epithelial cells [38], and is associated with ER positive status as well as favorable prognosis [39,40]. In our study, the subgroup with high (50% or above) GATA3 expression had the highest AR expression; not surprisingly, 17 of 19 BrM which were GATA3 high/AR positive were also HR+. 

The fact that our study shows a high proportion of breast cancer BrM are AR-positive using a biomarker predictive of response to anti-androgen therapies, supports the evaluation of AR-targeted therapies in breast cancer patients BrM. Table 3 summarizes 16 currently active clinical trials investigating AR-targeted therapies in advanced and metastatic breast cancer. It is encouraging that the majority (eleven of sixteen) of these trials include patients with controlled BrM. Six trials evaluate androgen agonist therapies (enobosarm, EP0062, EG017 and CR 1447); out of these trials, the majority (five of six trials) include patients with ER+/HER2− breast cancer, and two include patients with TNBC. Ten trials evaluate androgen antagonist therapies (enzalutamide, bicalutamide, and seviteronel); out of these trials, the majority (seven of ten trials) include patients with TNBC, five include patients with ER+/HER2− breast cancer, and only one includes patients with HER2+ breast cancer (and this trial excluded patients with BrM). Given that previous trials have shown the clinical response rates of antiandrogens in AR+ TNBC to be underwhelming, in the range of 19–35% [8,9,10,11,12], it is important that ongoing and future trials evaluate potential predictive biomarkers of response to optimize patient selection, as well as investigate combination therapies. 

Our study is limited by the small retrospective single centre design, and the fact that no anti-androgen therapies were prescribed to our patients. The small number of patients with matched primary and BrM tissue is another limitation. In the future, pre-clinical models of AR-positive BrM would be useful in validating these findings.

## 5. Conclusions

This study used a clinically validated antibody (SP107) to show in a 57-patient retrospective Canadian cohort that the majority of breast cancer BrM were AR positive (defined as AR ≥ 10% by IHC). Furthermore, AR expression was most frequently expressed in HER2+ BrM and least frequently expressed in TNBC BrM. These findings are comparable to AR expression previously reported in primary breast cancers. HER2+ BrM most frequently express AR, yet there are no active clinical trials evaluating AR-directed therapies in patients with HER2+ BrM; hence, additional trials for this population are warranted. While AR-targeted therapies may be most clinically useful in the treatment of TNBC, AR is expressed least frequently in TNBC BrM, suggesting a need for improved biomarkers to judiciously guide patient selection. Overall, our study shows that AR is a frequently expressed and promising intracranial molecular target, supporting the investigation of androgen-targeted therapies in the treatment of patients with breast cancer BrM.

## Figures and Tables

**Figure 1 cancers-15-02748-f001:**
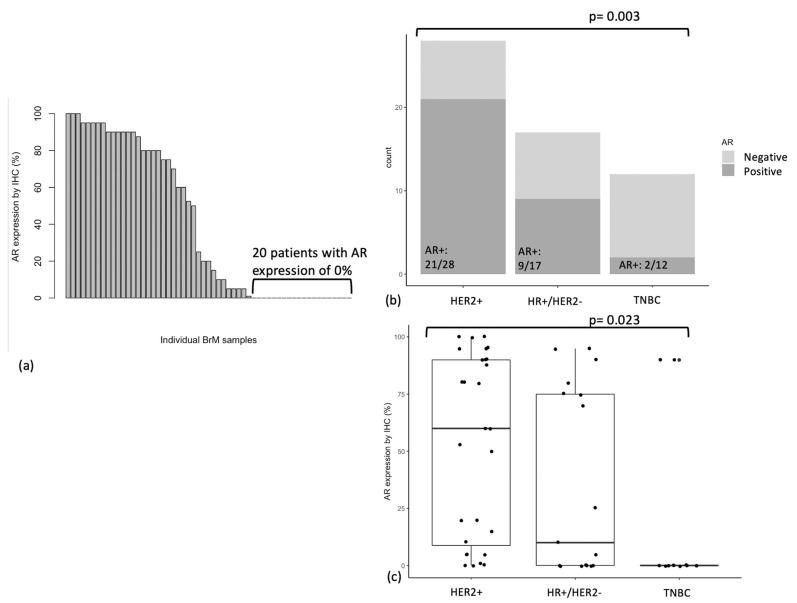
(**a**) AR expression (%) by IHC in breast cancer BrM (n = 57), (**b**) AR status (<10% or ≥10%) by breast cancer subtype, (**c**) AR expression (%) by breast cancer subtype.

**Figure 2 cancers-15-02748-f002:**
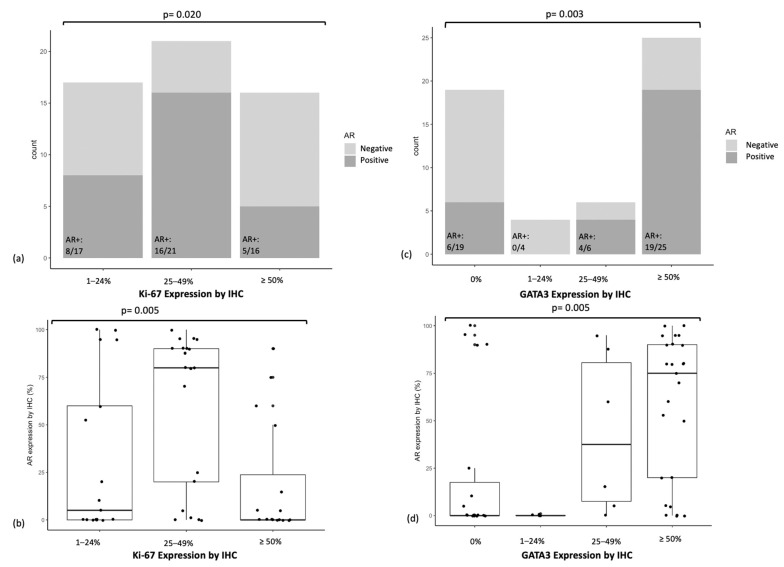
(**a**) AR status (<10% or ≥10%) in relation to Ki-67 expression, (**b**) AR expression (%) in relation to Ki-67 expression, (**c**) AR status (<10% or ≥10%) in relation to GATA3 expression, (**d**) AR expression (%) in relation to GATA3 expression.

**Figure 3 cancers-15-02748-f003:**
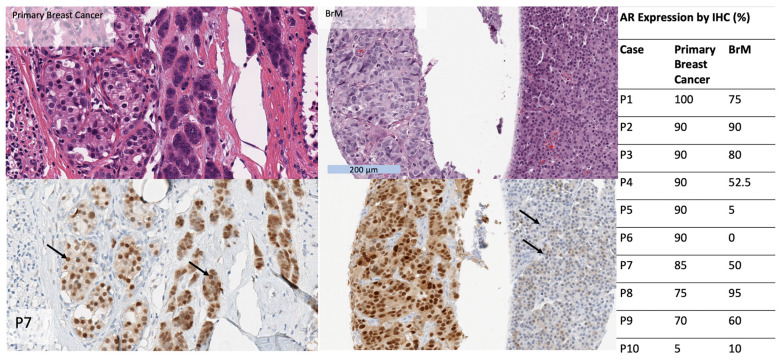
Table (**right**): AR expression by IHC (in %) in BrM compared to matched primary breast tumours, in a subset of 10 patients. Images show an example of a primary breast cancer and BrM from the same patient, hematoxylin and eosin (**top**) and SP107 IHC (**bottom**). The primary and BrM show morphologic heterogeneity with cells with lower nuclear grade and relatively abundant cytoplasm and cells with higher nuclear grade and scant cytoplasm. AR expression is diffusely positive in both components in the primary tumor but AR expression is diminished in the higher grade component in the BrM (arrows).

**Figure 4 cancers-15-02748-f004:**
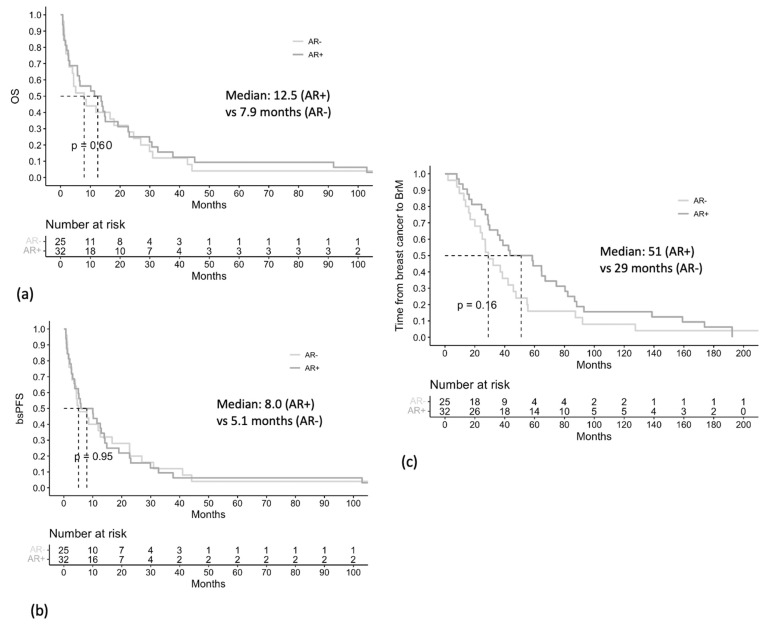
Overall survival (OS) (**a**), brain-specific progression-free survival (bs-PFS) (**b**), and time from diagnosis of breast cancer to diagnosis of BrM (**c**), stratified by BrM AR status (<10% vs. ≥10%).

**Table 1 cancers-15-02748-t001:** Review of active clinical trials evaluating androgen-targeted therapies in advanced and metastatic breast cancer. The search was conducted on 25 January 2023.

NCT Identifier	Trial Status	Phase	Patient Population	AR Status	Drug Category	Drug Class	Drug Name	Combination Drug(s)	Inclusion of BrM
NCT05673694	Not yet recruiting	1a/1b	Metastatic or recurrent ER+/HER2− breast cancer	AR positive	AR agonist	Non-steroidal androgen receptor agonist	EG017		Controlled BrM only
NCT02971761	Active, not recruiting	2	Metastatic TNBC	AR positive (≥50%)	AR agonist	Selective androgen receptor modulator	Enobosarm	Pembrolizumab	Controlled BrM only
NCT05573126	Not yet recruiting	1 and 2	Metastatic or locally advanced ER+/HER2− breast cancer	AR positive (≥30% using SP107)	AR agonist	Selective androgen receptor modulator	EP0062		Included
NCT04869943	Recruiting	3	Metastatic ER+/HER2− breast cancer	AR positive (≥40%)	AR agonist	Selective androgen receptor modulator	Enobosarm		Controlled BrM only
NCT05065411	Recruiting	3	Metastatic ER+/HER2− breast cancer	AR positive (≥40%)	AR agonist	Selective androgen receptor modulator	Enobosarm	abemaciclib	Controlled BrM only
NCT02067741	Active, not recruiting	2	Metastatic or locally advanced ER+/HER2− or TNBC	TNBC cases need to be AR+	AR agonist	Testosterone analogue	CR1447		Controlled BrM only
NCT01889238	Active, not recruiting	2	Advanced TNBC	AR positive	AR antagonist	Non-steroidal antiandrogen	Enzalutamide		Excluded
NCT05095207	Recruiting	1 and 2	Metastatic HER2−	AR positive (≥1%)	AR antagonist	Non-steroidal antiandrogen	Bicalutamide	Abemaciclib	Controlled BrM only
NCT03207529	Recruiting	1	Metastatic PTEN+, TNBC or HR+/HER2−, breast cancer	AR positive (≥1%)	AR antagonist	Non-steroidal antiandrogen	Enzalutamide	Alpelisib	Controlled BrM only
NCT02091960	Active, not recruiting	2	Locally advanced or metastatic HER2+ breast cancer	AR positive	AR antagonist	Non-steroidal antiandrogen	Enzalutamide	Traztuzumab	Excluded
NCT03090165	Recruiting	1 and 2	Metastatic TNBC	AR positive (≥10%)	AR antagonist	Non-steroidal antiandrogen	Bicalutamide	Ribociclib	Controlled BrM only
NCT02605486	Active, not recruiting	1 and 2	Metastatic TNBC	AR positive (≥1% by AR441)	AR antagonist	Non-steroidal antiandrogen	Bicalutamide	Palbociclib	Controlled BrM only
NCT03650894	Recruiting	2	Metastatic or locally advanced HER2− breast cancer	TNBC cases need to be AR+	AR antagonist	Non-steroidal antiandrogen	Bicalutamide	Nivolumab and Ipilimumab	Controlled BrM only
NCT02955394	Active, not recruiting	2	T2 or greater ER+/HER2− breast cancer	AR positive	AR antagonist	Non-steroidal antiandrogen	Enzalutamide	Fulvestrant	Excluded
NCT02007512	Active, not recruiting	2	Advanced ER+ (and/or PR+) HER2− breast cancer	No requirement	AR antagonist	Non-steroidal antiandrogen	Enzalutamide	Exemestane	Excluded
NCT04947189	Not yet recruiting	1 and 2	Metastatic TNBC	AR positive (>0% by IHC or gene classifier molecular testing)	AR antagonist	nonsteroidal CYP17A1 inhibitor	Seviteronel	Docetaxel	Controlled BrM only

**Table 2 cancers-15-02748-t002:** Patient and tumour characteristics.

Characteristic	*N* = 57
**Age at BrM diagnosis (years)**	
Median (IQR)	51.8 (14)
Range	32–85
**BrM subtype**	
Triple negative	12 (21%)
HER2+	28 (49%)
HR+/HER2−	17 (30%)
**Number of BrM**	
One	35 (61%)
More than 1	22 (39%)
**BrM size (cm)**	
Median (IQR)	3 (1.4)
Range	0.3–6.2
**BrM location**	
Frontal	15 (26%)
Parietal	13 (23%)
Temporal	3 (5%)
Cerebellar	25 (44%)
Occipital	1 (2%)
**BrM grade**	
1	16 (28%)
2	20 (35%)
3	15 (26%)
Unknown	6 (11%)
**Symptomatic BrM**	
Yes	50 (88%)
No	7 (12%)
**Sites of extra-cranial metastatic disease**	
Lung	17 (30%)
Liver	14 (25%)
Lymph node	11 (19%)
Bone	22 (39%)
Chest wall	2 (4%)
Other	4 (7%)
**Radiotherapy for BrM**	
Yes	51 (90%)
No	3 (5%)
Unknown	3 (5%)
**Systemic therapy for metastatic disease prior to BrM**	
Chemotherapy	9 (16%)
Trastuzumab-based treatment	10 (18%)
Endocrine therapy	4 (7%)
Unknown	34 (60%)
**Primary breast cancer subtype**	
Triple negative	4 (7%)
HER2+	16 (28%)
HR+/HER2−	12 (21%)
Unknown	25 (44%)
**Breast cancer stage at presentation**	
I	12(21%)
II	14 (25%)
III	9 (16%)
IV	1 (2%)
Unknown	21 (37%)

**Table 3 cancers-15-02748-t003:** Clinicopathological features stratified by BrM AR status (AR+ defined as AR expression by IHC ≥ 10%).

Characteristic	Patients	Number (%) of AR+ Cases	*p*
**Overall**	57	32 (56%)	
**BrM subtype**			0.003
Triple negative	12	2 (17%)	
HER2+	28	21 (75%)	
HR+/HER2−	17	9 (53%)	
**Ki-67 expression**			0.02
Low (1–24%)	17	8 (47%)	
Intermediate (25–49%)	21	16 (76%)	
High (≥50%)	16	5 (31%)	
Unknown	3	3 (100%)	
**GATA3 expression**			0.003
Negative (0%)	19	6 (32%)	
Low (1–24%)	4	0 (0%)	
Intermediate (25–49%)	6	4 (67%)	
High (≥50%)	25	19 (76%)	
Unknown	3	3 (100%)	
**Age at BrM (years)**			0.64
<50	23	14 (61%)	
≥50	33	17 (52%)	
Unknown	1	1 (100%)	
**BrM location**			0.95
Frontal	15	8 (53%)	
Parietal	13	8 (62%)	
Temporal	3	2 (67%)	
Cerebellar	25	14 (56%)	
Occipital	1	0 (0%)	
**BrM size (cm)**			0.25
<3.0	26	12 (46%	
≥3.0	27	19 (70%)	
Unknown	4	1 (25%)	
**BrM grade**			0.29
1	16	7 (44%)	
2	20	14 (70%)	
3	15	5 (33%)	
Unknown	6	6 (100%)	

## Data Availability

The data presented in this study are available in this article.

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
