# Peer review of "Androgen Receptor Is Expressed in the Majority of Breast Cancer Brain Metastases and Is Subtype-Dependent"

_cancers, 2023, doi:10.3390/cancers15102748_

Round 1
Reviewer 1 Report
This is a well written manuscript which deals with an interesting topic, namely the value of AR expresssion in BM of breast cancer patients.
I have some minor points to address:
- Table 3: Authors should give the date of the search (review of active clinical trials (by date.....)
- Table 1: I would recommend to write trastuzumab-based (instead of the drug name herceptin)
- table 1 after table 3 - please check order of tables!
- Table 2: p-value (check columns)
- please check also order of figures
Discussion:
Authors discussed AR immunohistochemistry (different antibodies), which is important in the context of AR determination.
I would recommend to delete the sentence line 295/296 (...has a strong trend towards...) because I think that results do not justify this sentence and it is misleading. Results are either statistically significant or not. Same for line 224-226.
Author Response
Thank you for these revisions. Please see attached word document for full response.
Response to reviewer 1:
This is a well written manuscript which deals with an interesting topic, namely the value of AR expression in BM of breast cancer patients.
I have some minor points to address:
- Table 3: Authors should give the date of the search (review of active clinical trials (by date.....)
We have updated the caption for Table 1 such that it now reads:
“Table 1. Review of active clinical trials evaluating androgen-targeted therapies in advanced and metastatic breast cancer. The search was conducted on January 25th, 2023.”
- Table 1: I would recommend to write trastuzumab-based (instead of the drug name herceptin)
We have changed the word “Herceptin” to “Trastuzumab”.
- table 1 after table 3 - please check order of tables!
We have corrected the order of tables
- Table 2: p-value (check columns)
We have added the correct p-value.
- please check also order of figures
We have corrected the order of figures
Discussion:
Authors discussed AR immunohistochemistry (different antibodies), which is important in the context of AR determination.
I would recommend to delete the sentence line 295/296 (...has a strong trend towards...) because I think that results do not justify this sentence and it is misleading. Results are either statistically significant or not. Same for line 224-226.
Thank you for this feedback. We deleted lines 295/296, lines 224-226 and subsequent two sentences. We have also have rephrased the section in the results that pertains to this. Taken together, we hope that these edits make it clear that this is not a statistically significant difference. The specific edits are listed below, with tracked changes:
“Our study found no significant association between AR expression and OS or bsPFS, which is not surprising, given that our cohort consisted of multiple different breast cancer subtypes, which inherently portend different prognoses, and the number of patients within each subtype was too small to draw meaningful conclusions. Additionally, no patients in our study received AR-targeted therapies.”
“This study used a clinically validated antibody (SP107) to show in a 57-patient retrospective Canadian cohort that the majority of breast cancer BrM were AR positive (defined as AR ≥10% by IHC). Furthermore, the frequency of AR expression was subtype dependent, and was most frequent in HER2+ BrM and least frequent in TNBC BrM. These findings are comparable to AR expression previously reported in primary breast cancers. HER2+ BrM most frequently express AR, yet there are no clinical trials evaluating AR-directed therapies in patients with HER2+ BrM, warranting additional trials for this population. While AR-targeted therapies may be most clinically useful in the treatment of TNBC, AR is expressed least frequently in TNBC BrM, suggesting a need for improved biomarkers to judiciously guide patient selection. Overall, our study shows that AR is a frequently expressed and promising intracranial molecular target, supporting the investigation of androgen-targeted therapies in the treatment of breast cancer BrM.”
“Patients with AR+ BrM had a median time of 51 months from diagnosis of breast cancer to diagnosis of BrM, compared to 29 months in patients with AR-negative BrM, although this was not statistically significant (51 vs 29 months, p= 0.16) (Fig 4. (c)).”

Reviewer 2 Report
Given the increasing incidence and therefore high relevance of brain metastases in breast cancer patients, the manuscript is of high clinical relevance despite the fact that currently not clinical approved agent is available targeting the androgen receptor. This limitation as well as the relatively low case number are adequately discussed.
The manuscript addresses the evaluation of a therapeutic target in brain metastases. As stated this is of potential clinical relevance. The topic is of clinical relevance. As far as I am aware, there is no publishes literature on this topic.
The limitations already mentioned by the authors (case number, no currently approved drug targeting AR in breast cancer) are discussed and are not possible to modify. As it is difficult to obtain a larger number of samples and this is not expected to change results, I would not ask for that.
The conclusions are consistent with the evidence and arguments presented and do they address the main question posed. The references are appropriate.
Author Response
Response to reviewer 2:
Thank you for your comments. As there were no changes/revisions requested, no changes were made in response to these comments.

Reviewer 3 Report
The authors evaluated the expression of AR in brain metastasis of breast cancer patients. The data was inconclusive due to limited number of patient specimens. The data that HER2-enriched tumors formed brain mets is a known phenomenon as HER2 enrichment increases brain mets of all cancers. The following are the comments to the authors.
- Analysis needs a larger patient number in order to gain more meaningful outcome.
- There needs to be more paired primary tumor data
- Review retrospective studies that have done RNA-seq on brain mets and look into the effects of AR mRNA expression and prognosis.
- Stain slides for proliferative and prognostic markers to assess if AR corresponds to prognosis in patients with brain mets.
- Need preclinical model(s) of AR-positive brain metastasis to validate the findings.
Author Response
Thank you for your revisions. We have done our best to address these revisions. Please refer to attached file and revised manuscript for full revisions.

Round 2
Reviewer 3 Report
None.